# Evolution of the Material Microstructures and Mechanical Properties of AA1100 Aluminum Alloy within a Complex Porthole Die during Extrusion

**DOI:** 10.3390/ma12010016

**Published:** 2018-12-20

**Authors:** Ding Tang, Wenli Fang, Xiaohui Fan, Tianxia Zou, Zihan Li, Huamiao Wang, Dayong Li, Yinghong Peng, Peidong Wu

**Affiliations:** 1State Key Laboratory of Mechanical System and Vibration, Shanghai Jiao Tong University, Shanghai 200240, China; tangding@sjtu.edu.cn (D.T.); Fangwll0@126.com (W.F.); Fanxhh0@126.com (X.F.); TianxiaZou0@126.com (T.Z.); Zihanllx@126.com (Z.L.); dylixx0@126.com (D.L.); yhpengx0@126.com (Y.P.); 2School of Mechanical Engineering, Shanghai Jiao Tong University, Shanghai 200240, China; 3Department of Mechanical Engineering, McMaster University, Hamilton, ON L4S 4L7, Canada; peidong@mcmaster.ca

**Keywords:** microchannel tube (MCT), extrusion, porthole die, microstructure evolution, VPSC model, flow line model

## Abstract

Microchannel tube (MCT) is widely employed in industry due to its excellent efficiency in heat transfer. An MCT is commonly produced through extrusion within a porthole die, where severe plastic deformation is inevitably involved. Moreover, the plastic deformation, which dramatically affects the final property of the MCT, varies significantly from location to location. In order to understand the development of the microstructure and its effect on the final property of the MCT, the viscoplastic self-consistent (VPSC) model, together with the finite element analysis and the flow line model, is employed in the current study. The flow line model is used to reproduce the local velocity gradient within the complex porthole die, while VPSC model is employed to predict the evolution of the microstructure accordingly. In addition, electron backscatter diffraction (EBSD) measurement and mechanical tests are used to characterize the evolution of the microstructure and the property of the MCT. The simulation results agree well with the corresponding experimental ones. The influence of the material’s flow line on the evolution of the orientation and morphology of the grains, and the property of the produced MCT are discussed in detail.

## 1. Introduction

Multi-port extrusion (MPE) tubes are widely applied in industry to transfer the substances of liquid, gas or slurries etc. As one of the MPE tubes, microchannel tube (MCT) exhibits excellent performance in heat transfer and have been employed more and more in heat exchangers. An MCT is commonly produced through a processing chain of extrusion, rolling and brazing [1,2,3,4]. The extrusion process, which involves severe plastic deformation, affects the final microstructure the most, though both rolling and brazing are important as well. The microstructure of aluminum alloys under thermal-mechanical loading have been extensively investigated [1,2,3,4,5,6], which eventually governs the mechanical properties of the MCTs. However, the MCT is produced by the porthole extrusion die with the specially designed mandrel [7]. The large extrusion ratio and the existence of the mandrel lead to assorted flow and severe deformation of the base material, and complicated evolution of the microstructure. The features of the microstructure developed under extrusion are inherited by the final products and affect their properties significantly. The authors’ group experimentally studied the development of the microstructure of aluminum MCTs during the porthole die extrusion [8]. We characterized the grain morphology and orientation in terms of electron backscattered diffraction (EBSD) and optical microscope (OM), where significant grain refinement and texture development were observed because of the severe plastic deformation. Moreover, remarkable variation of the microstructure along the through-thickness direction was obtained. We also realized that the quantitative interpretation of the development of the microstructure is hardly accomplished by merely experimental techniques. 

With the aid of constitutive modeling, it is believed that the underlying mechanisms associated with the development of the microstructure can be understood better. In recent years, the crystal plasticity methods have been extensively applied to understand physical mechanisms associated with large deformation [9,10,11,12,13,14,15]. Therefore, the viscoplastic self-consistent (VPSC) model, which is one of the most popular crystal plasticity models, is employed in the current work. The challenge of using VPSC model is that the material point will experience different deformation history at different location within the porthole die. Mayama et al. [16] found that the shear strain of the material is small around the central line, while that becomes relatively large at locations apart from the central line. The developed texture of AZ31 magnesium alloy under extrusion was also found to be different from location to location by Gall et al. [17]. 

As a consequence, in order to investigate the variation of microstructure of the material during manufacturing, the flow line method developed by Arruffat-Massion et al. [18] is employed in the current work to track the deformation history within the die. Experiments have demonstrated that the deformation history is more reasonably described by the metal flow line than a simple shear assumption [19,20,21]. The combination of the flow line method and the VPSC model enables the tracking of the evolution of the microstructure of aluminum alloy within the porthole die under extrusion. Reasonable agreement is obtained through the comparison between the simulations and experiments. The influence of strain paths on the inhomogeneity of microstructure is discussed in detail.

## 2. Experiments

### 2.1. Material

The material used in this study was an AA1100 commercial-purity aluminum alloy because of its excellent workability, weldability and corrosion resistance. The chemical composition of the alloy is given in Table 1. 

### 2.2. Extrusion

The extrusion experiment is performed using the porthole die shown in Figure 1b. Compared to a die for regular tubes (Figure 1a), porthole die has much more complicated flow pattern (Figure 1b). The structure of the mandrel and the produced tube are shown in Figure 1c. The aluminum alloy is divided by the porthole bridge into two nearly symmetrical material flows. The two material flows rejoin in the welding chamber and seam welded as they go through the mandrel teeth. The MCT is then formed when the material is pushed out of the die. The billet of 40 mm in diameter was lubricated with MoS_2_ paste and preheated to 480 °C, and then extruded with a ram speed of 5 mm/s. 

### 2.3. Microstructure Observation

In order to observe the development of the microstructure within the porthole die, the extrusion butt together with the die is quenched immediately after extrusion to preserve the grain structure during extrusion. The extrusion butt, together with the die, is cut into two halves through the extrusion direction (ED)-normal direction (ND) plane (Figure 2a). The ED-ND surface is prepared to characterize the microstructure on the longitudinal section. One half is further cut to obtain samples to characterize the microstructures in cross section (transverse direction (TD)-ND plane, Figure 2b). The obtained surfaces are further ground and polished using aluminum suspension and 0.02-μm colloidal silica suspension, then etched with modified Keller’s etchant for OM observation. For EBSD mapping, the surface is ground and polished by 2000# SiC paper, and then electropolished by LectroPol 5 (Struers (Shanghai) Ltd., Shanghai, China) with electrolyte A2. EBSD examination is carried out by NOVA NanoSEM 230 (FEI, Tustin, CA, USA) with Aztec HKL Max System with a step size of 2 μm. Along the metal flow line, seven typical positions of the extrusion butt are carefully scanned by EBSD. 

### 2.4. Mechanical Tests

In order to evaluate the influence of the microstructure on the mechanical behavior of the MCT tube, small specimens at different location of the tube are prepared for tensile tests. Corresponding tensile tests are conducted by an MTS CMT6103 universal testing machine manufactured by MTS Systems Corporation from San Diego, CA, USA. A Zwick/Roell Z020 universal testing machine (Ulm, Germany) is employed to conduct the compression tests with big specimen. 

## 3. Numerical Procedures

As shown in Figure 1, due to the geometry of the porthole die, the deformation gradient of the material is different from location to location. The numerical procedure is as following: (1) extrusion process was simulated by using finite element (FE) method to obtain the flow lines of the material; (2) the velocity gradient during the extrusion process is obtained by the flow line model [18]; and (3) the evolution of the microstructure is simulated by the VPSC model (Version 7d) based on the obtained velocity gradient [8,10]. 

### 3.1. Finite Element Analysis

The extrusion process is modeled in the commercial finite element (FE) software Deform 3D, which is widely used in simulation of hot extrusion process. The geometrical model associated with the porthole die is illustrated in Figure 3. The boundary conditions and material properties are chosen the same as our previous work [22], where the friction between the material and the inner surface of the die is accounted for in terms of the friction coefficient. The velocity field was extracted from FE simulation results, which is used to determine the velocity field of the material within the die. 

### 3.2. Flow Line Model

The flow line method has been proven to be an alternative effective method to quantitatively evaluate deformation histories in extrusion processes by directly describing flow paths of the material [19,23]. In the present work, an orthogonal Cartesian coordinate system is established with *x*, *y*, and *z*-axes along ED, ND and TD, respectively. Since the die dimension along TD is not changing, the material flow within the die (at the plane of z=0) can be described by a flow function:(1)ϕ(x,y)=y−f(x)=0,

An incompressible velocity field can be defined from the flow function ϕ as follows: (2)$$\{\begin{array}{c} v_{x}=\lambda \frac{\partial \phi }{\partial y}\\ v_{y}=-\lambda \frac{\partial \phi }{\partial x} \end{array}\;,$$
where $v_{x}$ and $v_{y}$ are the velocity components along ED and ND. λ is determined by the incoming velocity of the material $v_{0}$. Once the velocity field is obtained, the velocity gradient can be obtained by
(3)$$\{\begin{array}{c} \begin{array}{c} L_{xx}=\frac{\partial v_{x}}{\partial x}=\lambda \frac{\partial^{2}\phi }{\partial x\partial y}\\ L_{xy}=\frac{\partial v_{x}}{\partial y}=\lambda \frac{\partial^{2}\phi }{\partial y^{2}} \end{array}\\ \begin{array}{c} L_{yx}=\frac{\partial v_{y}}{\partial x}=-\lambda \frac{\partial^{2}\phi }{\partial x^{2}}\\ L_{yy}=\frac{\partial v_{y}}{\partial y}=-\lambda \frac{\partial^{2}\phi }{\partial x\partial y} \end{array} \end{array},$$

### 3.3. VPSC Model

Once the velocity gradient is determined, the VPSC model is employed to simulate the evolution of the microstructure accordingly. In the VPSC framework, the material is consisted of many grains. The aggregate of the grains is assumed as a homogenous equivalent medium (HEM), where each grain is an embedded ellipsoidal inclusion. The interaction between the HEM and the inclusion can be obtained by the Eshelby’s solution [24] in terms of the interaction tensor $M_{ijkl}$,
(4)$$\left({\dot{\varepsilon }}_{ij}-{\dot{E}}_{ij}\right)=-M_{ijkl}\left(\sigma_{kl}-\Sigma_{kl}\right).$$

The (plastic) strain rate (${\dot{\varepsilon }}_{ij}$) of each grain is accommodated by shear rate (${\dot{\gamma }}^{\alpha }$) on the slip system α,
(5)$${\dot{\varepsilon }}_{ij}=\sum_{\alpha =1}^{N}{\dot{\gamma }}^{\alpha }\left(s^{\alpha }m^{\alpha }+m^{\alpha }s^{\alpha }\right)/2,$$
where $s^{\alpha }$ and $m^{\alpha }$ are respectively the slip direction of normal of the slip plane, respectively. The shear strain rate of each slip system ${\dot{\gamma }}^{\alpha }$ follows the power law:(6)$${\dot{\gamma }}^{\alpha }={\dot{\gamma }}_{0}{\left|\frac{\tau^{\alpha }}{g^{\alpha }}\right|}^{n}sgn\left(\tau^{\alpha }\right),$$
where ${\dot{\gamma }}_{0}$ is the reference shear rate, n is the rate sensitivity parameter, $\tau^{\alpha }=s_{i}^{\alpha }\sigma_{ij}m_{j}^{\alpha }$ is the resolved shear stress (RSS), and $g^{\alpha }$ is the threshold stress. The evolution of $g^{\alpha }$ is given by
(7)$${\dot{g}}^{\alpha }=\frac{d{\hat{\tau }}^{\alpha }}{d\Gamma }\sum_{\beta }q^{\alpha \beta }\left|{\dot{\gamma }}^{\beta }\right|,$$
where $q^{\alpha \beta }$ is the matrix describing the latent hardening of the crystal and all populated by 1. ${\hat{\tau }}^{\alpha }$ is defined by an extended Voce law:(8)$$\hat{\tau }=\tau_{0}+\left(\tau_{1}+h_{1}\Gamma \right)\left(1-\exp\left(-\frac{h_{0}}{\tau_{1}}\Gamma \right)\right),$$
where Γ is the accumulated shear strain, $\tau_{0}$ and $\tau_{0}+\tau_{1}$, are the initial and back-extrapolated threshold stresses, $h_{0}$, and $h_{1}$ are the initial and asymptotic hardening rates, respectively. 

The grain refinement is considered in the VPSC model to crucially simulate the recrystallization. The shape of the ellipsoidal inclusion is characterized by three principal axes a, b, c with a≥b≥c, which evolve with deformation. When the length ratio of *a*/*c* the ellipsoidal grain reaches a critical value R, the grain will be refined to two grains with the dimension of a/2, b, c. The crystallographic orientation of the two grains remain the same as prior to refinement. 

## 4. Results and Discussion

### 4.1. Velocity Gradient Determination

The extrusion process is simulated by pushing the base material through the die. The model parameters are listed in Table 2. The velocity field and the effective strain field simulated by the Deform 3D are shown in Figure 4. As expected, heterogenous deformation, especially strong gradient along ND has been obtained. Three flow lines are extracted from the velocity field at locations close to the bridge, close to the top surface of the die, and in between. The extracted flow lines are plotted together with the strain rate field in Figure 5a. Again, strong heterogenous strain rate field is gained. We found that the flow lines extracted from the FE analysis can be well described by 4 sectioned polynomial functions with maximum degree of 4. Therefore, the function f(x) in Equation (1) can be expressed as:(9)$$f^{i}\left(x\right)=\sum_{k=0}^{k=4}a_{k}^{i}x^{k}\;x\in \left[x_{i-1},x_{i}\right],$$
where $a_{k}^{i}$ is the polynomial coefficient of the sectional function within the *i*^th^ zone of $\left[x_{i-1},x_{i}\right]$. The entire x range of the porthole die is composed of the four zones (i.e., *i* = 1, 2, 3 and 4). The polynomial fittings are shown in Figure 5b and the obtained polynomial coefficients are listed in Table 3. 

Figure 6 shows the variations of the grain size and orientation along the flow direction in the porthole extrusion die. In the inlet section, most of the grains are equiaxed. It also can be seen that the material undergoes shear deformation because of the difference of the flow speed from the center of the metal flow to the boundaries. As the metal enters the seam welding chamber, owing to the narrowed die channel, the shearing deformation becomes more significant. The grains are elongated and align towards the material flow line. Grain size of the material in dead zone is much larger than that in the other positions. However, dead zone induces large friction, and therefore large shear deformation, to the flow material in the vicinity. As a consequence, finer grains are observed in the EBSD maps at A6, which is closest to the dead zone. Together with the grain topology evolution, the orientations of the grains also evolve simultaneously. The two scanning results closest to the end of extrusion die (A6 and A7) along the extrusion direction show clear flow lines composed of similar orientation of textures. 

Figure 7 shows the variations of the microstructure of the cross section (ND-TD plane) at points A2, A4 and A6 within the porthole extrusion die (Figure 6). Obviously, equiaxed grains are observed in the three observation sections, which reflects that the deformation is not varying significantly within the ED-TD plane. 

### 4.2. Parameters Calibration

The initial texture at positions of A1, B1 and C1 for corresponding flow lines are depicted in Figure 8. The obtained initial textures are discretized into 1000 grains for the VPSC simulations. Uniaxial compression test at temperature of 450 °C is performed to characterize the mechanical behavior of AA1100. The experimental stress strain curves at strain rates of 0.1, 1 and 10 s^−1^ are fitted by the VPSC model to determine the model parameters. The fitting is shown in Figure 9, where good agreement is obtained, and the obtained parameters are listed in Table 4. 

### 4.3. Texture Evolution

With previous flow line model and calibrated hardening parameters, the texture evolution of aluminum, grain size and morphology development along the extrusion flow line are simulated, which can be used for analyzing rotation of grain orientation and mechanism of grain refinement in a sophisticated path.

The velocity gradient from Equation (3) can be decomposed into symmetric and asymmetric parts. The asymmetric part governs the rigid rotation of the material point within the die. Correspondingly, a co-rotational coordinate is established following the path of flowing particles to eliminate the influence of rigid rotation and better understand the effect of shear strain on texture evolution. The subsequent results reveal that large shear deformation is accompanied with the large rotation. The textures with respect to different positions along the extrusion direction are calculated and transformed onto corresponding normal plane of the flow lines. Figure 10, Figure 11 and Figure 12 present the textures at seven positions along each flow line to illustrate the evolution of the texture. 

Figure 10 shows the experimental and simulated {001} and {111} pole figures evolution on flow line A which is the outermost flow line. As can be seen, shear is not obvious in steady state deformation zone from A1 to A3, and shear plane doesn’t rotate significantly. However, the rotation of shear plane *x*’-*y*’ becomes relatively large during aluminum flows from A3 to A4, which can be attributed to the change of friction in aluminum flow since the first dead zone appear. The difference in friction relationship between the flowing material and the die, as well as between the flowing material and the stationary material in dead zone, results in counterclockwise rotation of the shear plane. Slight clockwise rotation happens in shear plane from A4 to A5. While, shear plane rotates counterclockwise again because of obstruction of aluminum in the second dead zone from A5 to A6. Clockwise rotation takes place from A6 to A7. Contrast to experiment result on flow line A, slight deviation between the experimental and numerical results can be found. The deviation takes place in position A3 and A5, where the interaction between flowing aluminum and stationary material in dead zone begins to occur. Because grain size of stationary material in dead zone is far greater than that in flowing aluminum, the obstruction between stationary material in dead zone and flowing aluminum is greater than the adhesive friction between die and aluminum, which results in clockwise rotation of the shear plane.

Compared with the results in line A, shear is not obvious in steady state deformation zone from B2 to B3 in line B (Figure 11). Shear plane *x*’-*y*’ rotates clock wisely when aluminum flows from position B1 to B2, B3 to B4, B4 to B5, and B5 to B6. Finally, at the die export position B7, slight counterclockwise rotation takes place. Material points on flow line B are furthest from the bridge and the external die, which makes the influence of friction negligible. 

Figure 12 shows {001} and {111} pole figures evolution on flow line C, which is the innermost flow line. Because of 120° corner transition design on die diversion bridge, shear plane *x*’-*y*’ rotate clockwise when aluminum flows from position C1 to C2. The shear plane is nearly not rotating in steady state deformation zone from C2 to C3. Counterclockwise rotation in shear plane takes place when aluminum flows into severe shear deformation zone from C3 to C4. However, shear plane rotates clockwise from C4 to C5 and C6 in severe shear deformation zone. Shear plane rotates counterclockwise from C6 to C7 in seam-welding chamber. In contrast to the experimental results, the simulated orientation of the grains keeps consistent with respect to deformation. However, on position C7, due to the joining of the two material flows, the status with high hydrostatic pressure produces a cubic orientation, which is not accounted for by the VPSC model. 

Through analysis for rotation law of shear plane above, the magnitude of shear on flow lines A, B, and C is different from each other. The main reasons are as follows: on flow line B, the essential factor for texture orientation change is the plastic flow of material; on flow line A, the change of shear plane is affected by dead zone; on flow line C, the rotation of shear plane is restrained by the bridge structure. Therefore, the methodology of combining the flow line model and the VPSC model can accurately describe the development of grain orientation in complex material flow process. 

### 4.4. Grain Shape and Size Development 

The influences of deformation history on microstructure evolution are investigated. The microstructures at different extruded positions are examined by EBSD and the average grain size is calculated. Figure 13 shows that the evolutions of grain size and morphology on flow lines A, B and C. The position (i.e., deformation history) shows a significant effect on the microstructure evolution. 

From position of A3 to A4 in line A, a dramatic increase in the long axis of grains is observed. The reason is the severe shear, which is caused by the strong friction between the material and the die, makes grains severely stretched. Grain are further stretched and refined repeatedly from position A4 to A5, which results in a fluctuation of grain size. The rotation of shear plane makes grain further refined from position A5 to A6. Short axis of grain reaches saturated value because of the nearly uniform deformation. The aspect ratio shares a similar law with long axis, where dramatic fluctuation is also obtained from position A4 to A7. 

Grain size and morphology development on flow line B is different from flow line A. Grain morphology has not changed much in steady state deformation zone. At stage of severe plastic deformation (B4 to B5), grain refinement doesn’t occur because the gradient of the deformation across thickness is relatively low.

For the flow line C, grain size and morphology keep basically the same in steady state deformation zone and only shows slight increasing. Grain size decrease dramatically before reaching position C6, where a relatively large shear deformation makes significant grain refinement.

### 4.5. Mechanical Property across the Thickness of MCT

The specimens were prepared at different regions with respect to the three flow lines in the longitude section of the MCT tube, with the dimension of 0.2 mm × 0.2 mm × 10 mm (Figure 14a). The test for each flow line was repeated at least three times to ensure its reliability. The measured results are compared in Figure 14b, where the specimen at flow line B has the highest stress strain curve, the one at flow line A has the lowest, while the one at flow line C is in between. These different mechanical behaviors are associated with both the developed textures and grain morphology shown in Figure 10, Figure 11, Figure 12 and Figure 13. Figure 14c plots the ultimate strength together with the average grain size, where specimen with finer grains exhibits higher ultimate strength. This finding agrees well with Hall-Petch effect of the grain size strengthening.

In summary, by analysis of grain size and morphology development on different flow line, it is found that the position of grain refinement and degree of grain morphology evolution have a close relationship with the rotation of shear plane. The way of shear change decides the position where grain is stretched and refined. All these evolutions are well captured by current model. 

## 5. Conclusions

In the current work, the VPSC model with grain refinement is employed, based on the calculated strain rates from the flow line model, to examine the evolution and distribution of texture and grain size within the extrusion die. Results of the simulation agree well with EBSD observations. The influence of flow lines on the evolution of texture and grain morphology is discussed. Based on both the experimental and numerical results, the following conclusions can be drawn:(1)Large deformation is occurred under extrusion with a porthole die. Both the velocity and effective strain fields reveal that the base material undergoes very heterogenous and severe deformation. (2)The FE simulation can accurately reproduce the flow line of the material within the porthole die. The velocity gradient within the die can be obtained by the flow line model. In conjunction with the crystal plasticity model of VPSC, the evolution of the microstructure can be well reproduced.(3)The rotation of shear plane causes the texture change during material flow. The different rotation of shear plane attributes to the difference of texture evolution law on three flow lines. Because of mold structure and change of friction condition, the rotation of shear plane in deformation process performs alternation of clockwise and counterclockwise on flow line A and C. For flow line B, in dramatic plastic deformation zone, the shear plane keeps clockwise rotation, except counterclockwise rotation at the exit of mold.(4)There is a close relationship between grain refinement and shear effect. The grain refinement characteristics vary with different shear effect of aluminum on different flow line. Polycrystalline plasticity model predicts the trend of grain size and morphology development on different flow line, especially the process that grain is stretched and, to a certain extent, refined under severe deformation. (5)Because of the different microstructure developed, aluminum alloys in the produced tube exhibit different mechanical behaviors across the thickness from location to location. Good correlation to the Hall-Petch effect is obtained, where aluminum alloy with finer grains generally shows higher strength. 

## Figures and Tables

**Figure 1 materials-12-00016-f001:**
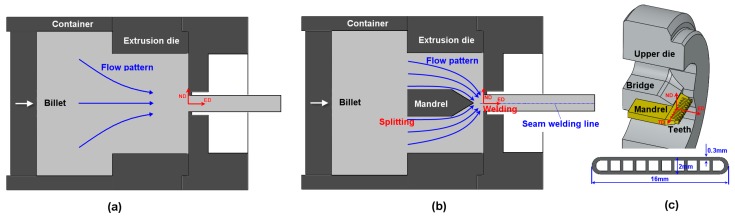
Schematic representations of (**a**) the regular die, (**b**) the porthole die, and (**c**) the mandrel structure and tube section.

**Figure 2 materials-12-00016-f002:**
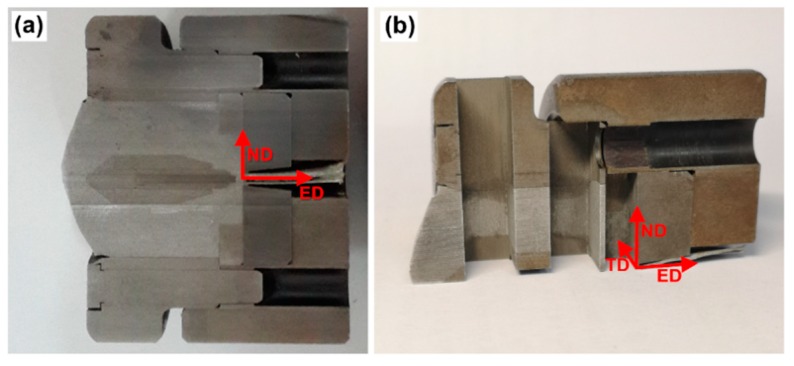
The observation section of the metal flow: (**a**) longitudinal section, and (**b**) cross section.

**Figure 3 materials-12-00016-f003:**
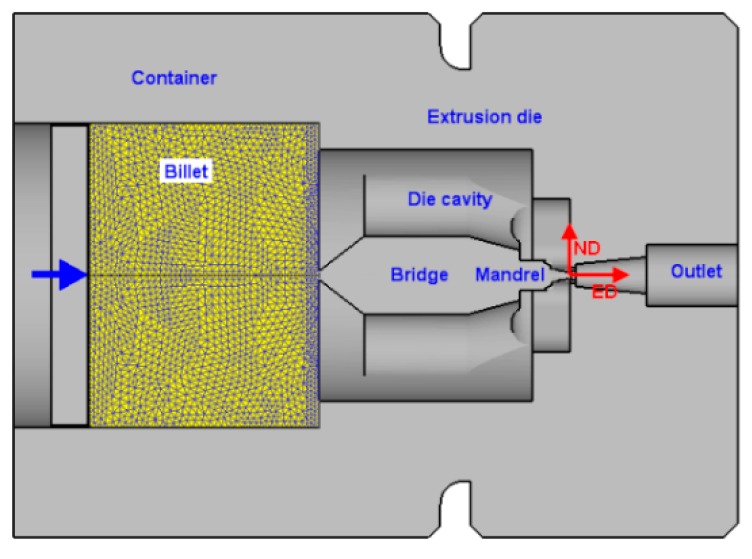
The finite element model generated by Deform 3D.

**Figure 4 materials-12-00016-f004:**
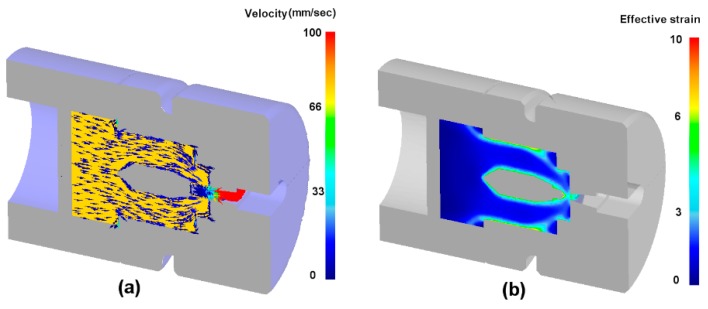
FE simulation results: (**a**) velocity field and (**b**) effective strain field.

**Figure 5 materials-12-00016-f005:**
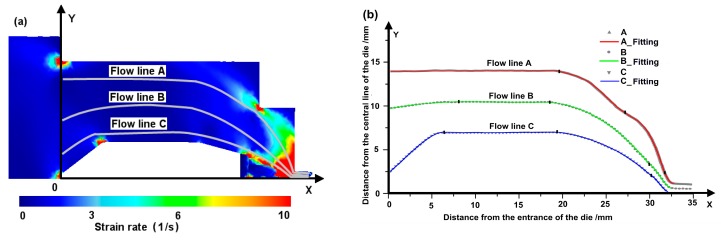
Flow lines of the aluminum alloy under extrusion within the porthole die: (**a**) FE simulation; (**b**) extracted flow lines and polynomial fitting.

**Figure 6 materials-12-00016-f006:**
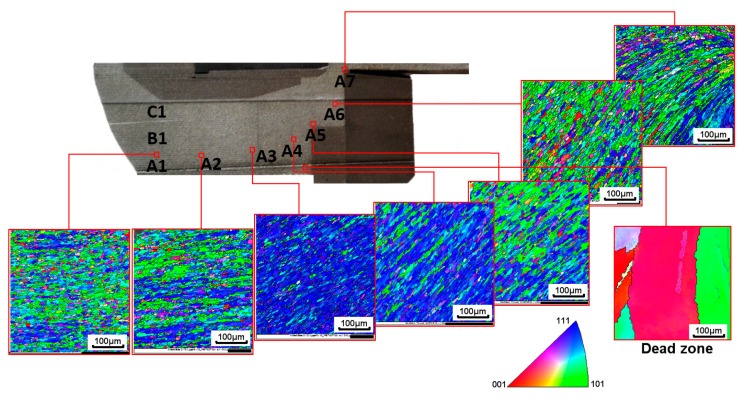
Microstructure evolution in the longitudinal section along the flow line A characterized by EBSD.

**Figure 7 materials-12-00016-f007:**
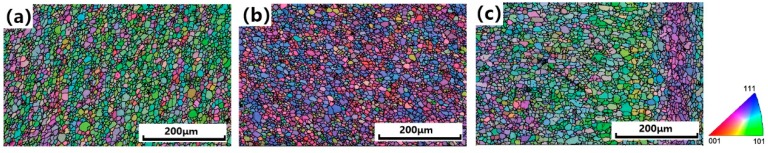
Microstructure evolution in the cross section (ND-TD plane) characterized by EBSD: (**a**) point A2; (**b**) point A4; (**c**) point A6.

**Figure 8 materials-12-00016-f008:**
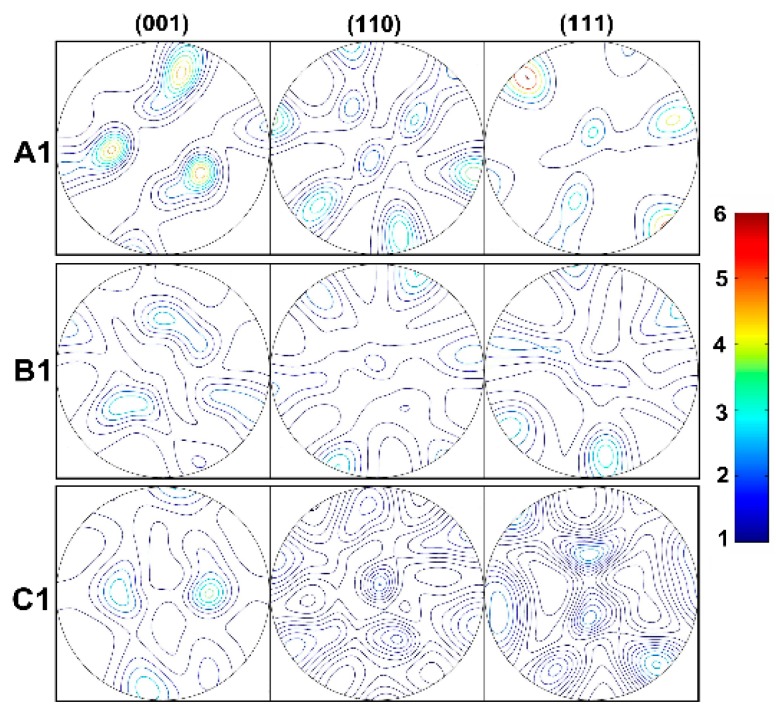
Initial textures obtained by EBSD at the beginning of the three flow lines.

**Figure 9 materials-12-00016-f009:**
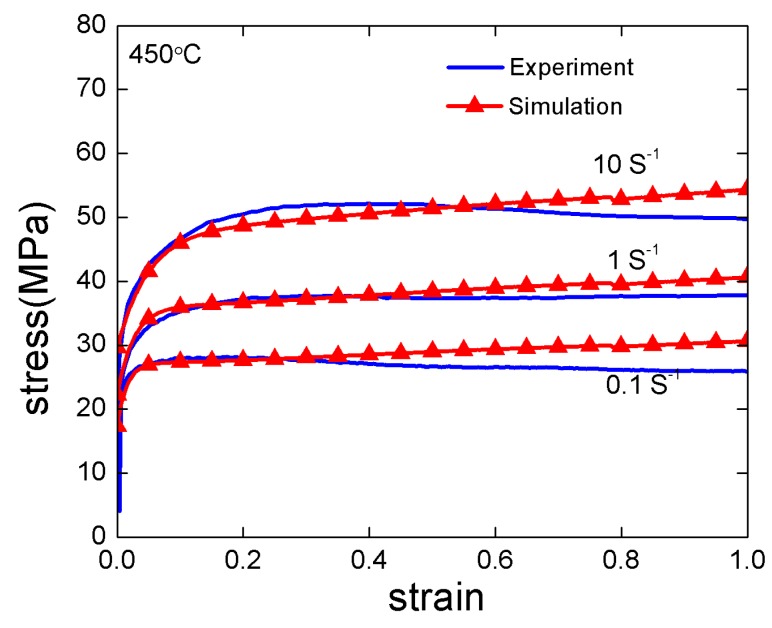
Experimental and simulated stress strain curves at different strain rates under compression.

**Figure 10 materials-12-00016-f010:**
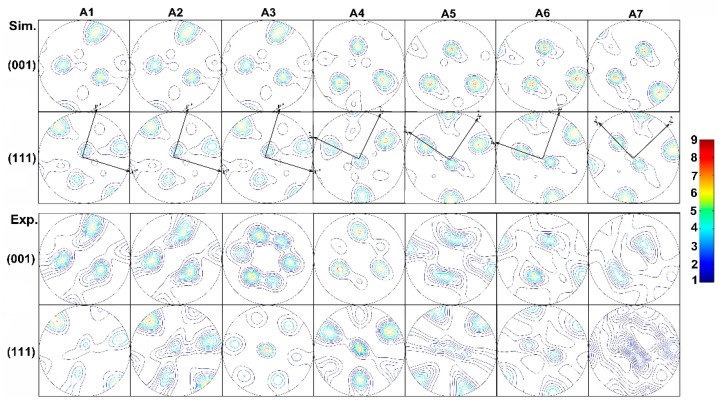
Experimental and Simulated {001} and {111} pole figures along the flow line A.

**Figure 11 materials-12-00016-f011:**
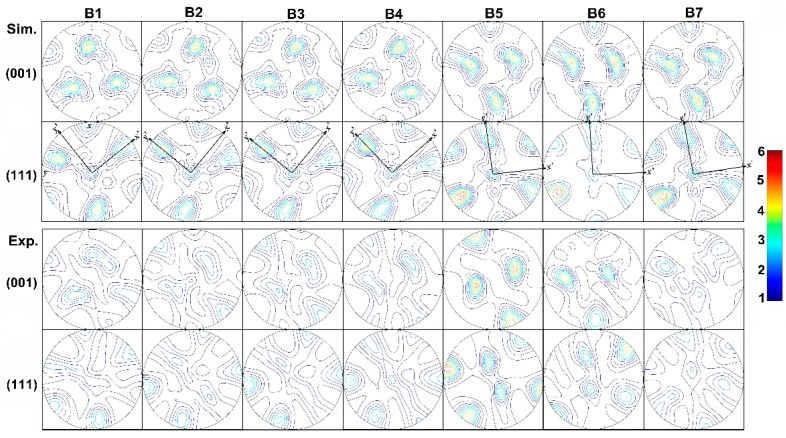
Experimental and Simulated {001} and {111} pole figures along the flow line B.

**Figure 12 materials-12-00016-f012:**
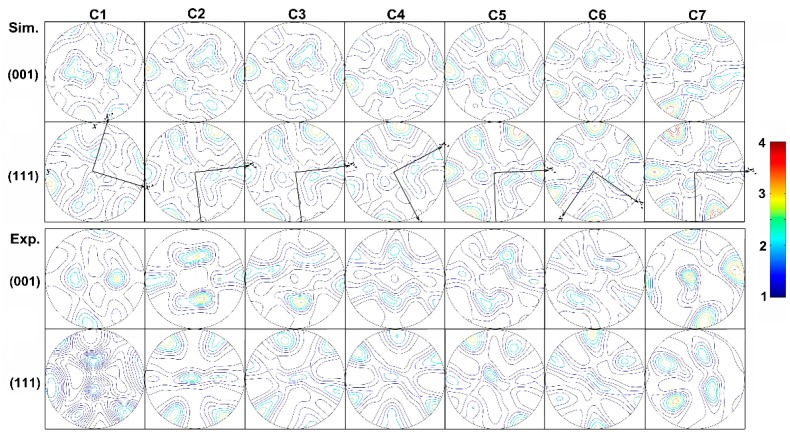
Experimental and Simulated {001} and {111} pole figures along the flow line C.

**Figure 13 materials-12-00016-f013:**
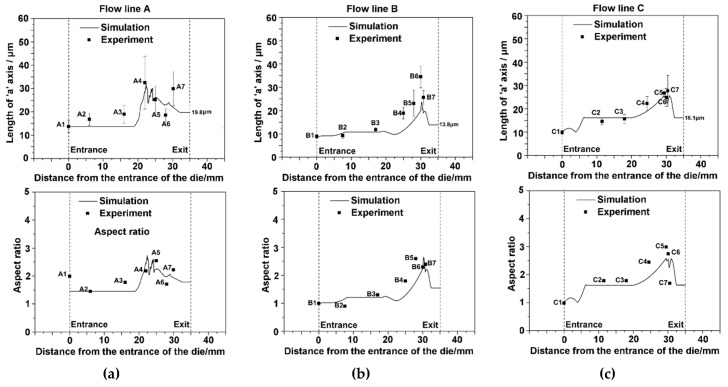
Experiment and simulation of the grain morphology in terms of grain size and aspect ratio associated with the (**a**) flow line A; (**b**) flow line B; and (**c**) flow line C.

**Figure 14 materials-12-00016-f014:**
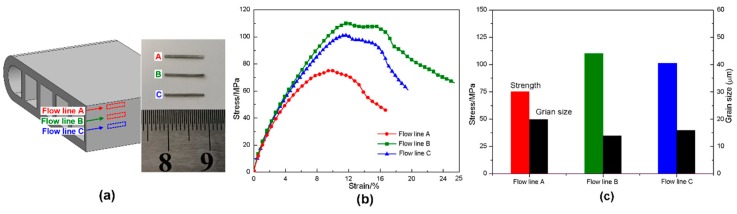
Tensile tests on specimens at different flow lines: (**a**) sampling position; (**b**) engineering stress-strain curves; and (**c**) ultimate strength.

**Table 1 materials-12-00016-t001:** Chemical composition of the aluminum alloy (AA1100).

AA1100	Al	Si	Fe	Cu	Mn	Zn	Others
wt. %	99.39	0.09	0.23	0.18	0.0005	0.0035	0.106

**Table 2 materials-12-00016-t002:** Billet material property and processing parameters in extrusion process.

Parameters	Values
Container inner diameter (mm)	40
Extrusion ratio	150
Ram speed (mm/s)	5
Billet temperature (°C)	480
Die temperature (°C)	450
Force on dummy block (kN)	460

**Table 3 materials-12-00016-t003:** Polynomial coefficients of the flow lines shown in Equation (8).

Flow Line	Zone	$a_{0}$	$a_{1}$	$a_{2}$	$a_{3}$	$a_{4}$
upper line	[0, 19.7]	13.6	2.4 × 10^−3^	0	0	0
	[19.7, 27.3]	804.1	−141.9	9.5	−2.8 × 10^−1^	3.1 × 10^−3^
	[27.3, 31.7]	13,405	1799	90.3	−2	1.7 × 10^−2^
	[31.7, 32.8]	1264.8	−77	1.2	0	0
middle line	[0, 8.1]	9.3002	0.207	−0.019	0.001	−3 × 10^−5^
	[8.1, 18.4]	10.2	−5 × 10^−3^	0	0	0
	[18.4, 30.0]	3.9	0.5	5 × 10^−3^	−7.4 × 10^−4^	0
	[30.0, 32.4]	−5388.8	521.7	−16.8	1.8 × 10^−1^	0
lower line	[0, 6.4]	2.2	0.76	8 × 10^−2^	−1.4 × 10^−2^	0
	[6.4, 19.3]	6.7	−4.8 × 10^−3^	0	0	0
	[19.3, 30.1]	8.3	−0.36	3.1 × 10^−2^	−8.82214 × 10^−4^	0
	[30.1, 32.6]	−6006.2	581.5	−18.7	0.2	0

**Table 4 materials-12-00016-t004:** Voce hardening parameters.

$\tau_{0}$	$\tau_{1}$	$h_{0}$	$h_{1}$	n	R
10	5.9	39	0	8.5	5

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
