# Peer review of "Evolution of the Material Microstructures and Mechanical Properties of AA1100 Aluminum Alloy within a Complex Porthole Die during Extrusion"

_materials, 2018, doi:10.3390/ma12010016_

Reviewer 1 Report

Dear authors,

 Your work shows a good agreement of simulation and experiment, is very well structured with the results well-presented and the conclusions deriving from both experimental and simulation work performed. Congratulations. This is a very nice work that deserves attention and must, thus, be published.

Please include the step size of the EBSD measurement as this is important to support the accuracy of the measurement especially due to the very high deformation applied, which creates a very refined grain structure.

Additionally, I suggest to discuss somehow more extensively what happens in the vicinity of the dead flow zone. In deed a change in friction occurs and this influences both the progress of the extrusion process, but also influences the microstructure. This can be better shown by the EBSD results. However, the manuscript as it stands, is clear enough and an experienced reader wouldn’t need too much in depth analysis regarding this issue.

Herewith, I accept the publication of the manuscript in its current form.

Sincerely,

The reviewer

Author Response

please find author's response in the attached file.

Reviewer 2 Report

The article titled Evolution of the Material Microstructures and Mechanical Properties of A1100 Aluminum Alloy within a Complex Porthole Die during Extrusion represents a valuable contribution to understanding the microstructure and mechanical properties of aluminum micro channel tubes (MCT) produced by extrusion. The influence of the material's flow line on the evolution of the orientation and morphology of the grains, and the properties of the produced MCT was investigated by simulation models and experimental studies that have showed a high correlation between the results. The results were discussed in detail.

In terms of improving the quality of article, the authors are sugested to do the following:

 1. Line number 58: When labeling AZ31 alloys, it should have been mentioned that this is a magnesium alloy.

2. Line number 76: On the Figure 1, the signatures below each image should be deleated as they are already contained in the image name. In Figures 1(a), 1(b) and 1(c), it is necessary to emphasize the Cartesian coordinate system as well as the axis marks.

3. Line numbers 75 - 85: When specifying the extrusion parameters in Section 2.2 Extrusion, it would be desirable to adduce the value of applied force or pressure. The symbol of MoS2 paste should be corrected (not MoS2 but MoS2)

4. Line numbers 100 - 103: In which measuring range was the tensile test performed since it has extremely small forces of only a few N if considered the results of the measured stresses and the cross-sectional dimension of the test samples are only 0.2 mm x 0.2 mm. The MTS CMT6103 type label should preferably include the name of the manufacturer and the country of origin, and the term "testing equipment" should be renamed "universal testing machine".

5. Line number 135: The point in the middle of the sentence should be deleted.

6. Line number 146 and 149: What is the symbol "Γ" mentioned in Equations 6 and 7?

7. Line numbers 159 - 160: Why are the model parameters listed in Table 2 used for the FEM simulation of the extrusion process different from the actual extrusion parameters mentioned in Section 2.2 Extrusion?

8. Line number 173: The word "ith" should be corrected in "ith".

9. Line numbers 180 - 181: Authors should clarify the corresponding coordinates of the zones mentioned in Table 3, for which polynomial coefficients are determined. For X coordinate, it can still be assumed to indicate the distance from the entrance of the die as shown in Figure 5b, while the Y coordinate is not in the scale shown in this figure.

10. Line numbers 205 - 207: It is mentioned that uniaxial compression test was carried out at a temperature of 450°C but no device was specified anywhere. This should be emphasised in Chapter 2.4 Mechanical tests.

11. Line number 212: What do mean the sizes "Θ0" and "Θ1"   mentioned in Table 4?

12. Line number 232: Incorrect number of image. (Figure 9 should be replaced with Figure 10)

13. Line number 232: It is necessary to correct the label of the measurement unit (not "mmm" but "mm").

14. Line number 300: When the authors in Figure 14 (b) give the results of the tensile tests, it would be desirable to mention the number of repetitions or the number of samples tested on each of the three flow lines. Is there one or more samples on each flow line?

15. Line number 391: It should be replace the marks of the journal number and the year of issue (consistent with other literary sources should be written "214 (2014)").

Author Response

(The authors gave the same response as above.)

Reviewer 3 Report

L 137-138 : The authors derived the interaction between the HEM and the inclusion obtained by the Eshelby’s solution [24]. Please indicate the place or equations where you used in Ref.24 and write that in this article. 

As for Fig. 5 (b), 8, 10, 11, 12 : The characters (words ) in each figures is small. Please write with larger point size in order to read easily. Especially, in Fig. 8, 10, 11, 12, the printed characters are rough.

Author Response

Please find author's response in the attached file.
